# A Decision Support System for Hospital Configurations in Construction Projects

**Pia Schönbeck** , **Malin Löfsjögård and Anders Ansell** *

Department of Civil and Architectural Engineering, KTH Royal Institute of Technology, 10044 Stockholm, Sweden
* Correspondence: ansell@kth.se

**Abstract:** Hospitals are complex buildings and their functionality is essential for patient healthcare. Definition, verification and implementation of accurate configuration information during construction projects are therefore essential. The purpose of this study was to develop a decision support system by establishing a value chain of configuration information with an end-to-end perspective. The approach of this study was explorative, investigating how building data can support construction projects in making hospital configuration decisions. A literature review provided a knowledge base about the configuration decision process flow, which determined the prerequisites for the proposed data and model management. Exchange and relationships of required building data were ensured by using Industry Foundation Classes (IFC) and a database model, respectively. The results show that using building model data for configuration decision support is feasible. A case study compared data exchanged in three construction projects of Magnetic Resonance Imaging rooms to those identified in the decision support system. Operational gaps regarding data exchange in the studied cases indicate what changes are required in current data collection and management. The contribution of this study is filling a research gap regarding end-to-end information management to support hospital configuration decisions in construction projects.

**Keywords:** decision support system; data driven processes; configuration information; building model data; digitalisation; hospital buildings

## 1. Introduction

Buildings have complex configurations containing spaces with different requirements regarding functionality. The management of configuration information has a lifecycle perspective [1] and can create value even after demolition by enabling knowledge transfer for future purposes [2]. The use of digital models during construction projects is often focused on design [3]. However, the end-to-end process for configuration control entails multiple areas, such as functional requirements, design solutions, construction methods, verification and change management. Lifecycle perspectives require the horizontal integration of end-to-end solutions across organisations and industries [4]. Hospital configurations require multiple complex solutions to enable patient care. Several organisations and individuals must collaborate to accomplish adequate configurations [5]. Decisions based on data analysis can increase deliveries of hospitals with the intended functionality [6].

The scope of this study is to provide new knowledge on how building data can support hospital configuration decisions. First, a flow chart for configuration decisions during construction projects was established, based on the ISO 10007 guidelines for configuration management [1] and a literature review. Second, an entity-relationship (ER) model established relations between decision entities to support configuration decisions. For each entity, relevant schemas from Industry Foundation Classes (IFC) were identified to ensure possibilities for data exchange between involved actors. The final part was predicting operational gaps by comparing differences between IFC schemas from the proposed ER model with those of hospital construction projects delivering rooms for Magnetic Resonance

Imaging (MRI) equipment. Construction research often investigates project issues, such as cost, time and processes. However, several of these issues are connected with inadequate configuration management and broken information chains [7]. There are studies addressing different aspects of information management in construction projects, such as functional requirement analysis, computerised designs and industrial production methods. However, few address the end-to-end perspective with a value chain regardless of actors and phases [8]. In configuration information management, this phase transcending perspective is a prerequisite for controlling end-product performance. The practical implications of this study are data-based support of configuration decisions that can improve the accuracy of deliveries and standardise the realisation.

### 1.1. Configuration Decisions in Construction Projects

Construction projects receive, maintain and deliver building configuration information to the property owner. The first step is collecting operational functionality requirements [9]. Transformation of these requirements to measurable acceptance values enables their verification during design and construction [7]. All required functions constitute the baseline configuration, which is the boundary conditions for the entire construction process [1]. Insufficient client involvement in determining function acceptance values results in incomplete baseline configurations [10], which leaves designers with unclear boundaries for their solutions [11]. The design specifications should clearly define the product to be built, but these often contain errors and omissions [12]. This results in unachievable design solutions, causing inefficiency and an increased number of changes during the realisation. Designer and contractor collaboration can ensure feasible production methods for each design solution [13]. The verification of the end-product performance during construction projects is mainly performed as built, resulting in rework [14]. Continuous control of function fulfilment increases the possibility of delivering an adequate end-product (Ding et al., 2017). Analyses of how configuration changes affect the end-product functionality are essential for an accurate delivery [15].

Configuration information aims to describe product characteristics in real time through all phases of a product's lifecycle [1]. Therefore, the configuration information must be reliable and correct at all times [16]. Reliable configuration information reduces the risk of errors and reworks [14]. Transparency and accessibility to information are prerequisites for configuration control, which sequential management in construction projects often impedes. Currently, the digital information describing a building configuration is part of the handover, together with the as-built construction. Incorrect information in the end-to-end process of construction projects may result in inadequate deliverables affecting end-user operations [9]. Configuration information ensures the fulfilment of functions throughout the entire construction project. Evaluations of how production methods and design solutions fulfil functions enable standardisation and optimisation of building configurations, as shown in Figure 1. Configuration information management extends beyond a building's life cycle by transferring knowledge to future projects and enabling optimisations [17]. Therefore, a horizontal approach is required to establish value chains independently of actors or operations [18].

### 1.2. Hospital Configurations

For hospital buildings, control over configurations is especially important since insufficient functionality may affect patient healthcare. Construction projects have problems with inadequate quality, cost overruns and delays [19]. These issues relate to insufficient configuration control, such as deficiently defined deliverables [20], inadequate feasibility analysis [21] and uncontrolled changes [15]. Hospital configurations are complex since they contain facilities with specific requirements on functionality. However, few studies investigate how configurations fulfil different functional requirements [22,23], such as post-occupancy evaluations of which design solutions enable end-user operations [24].

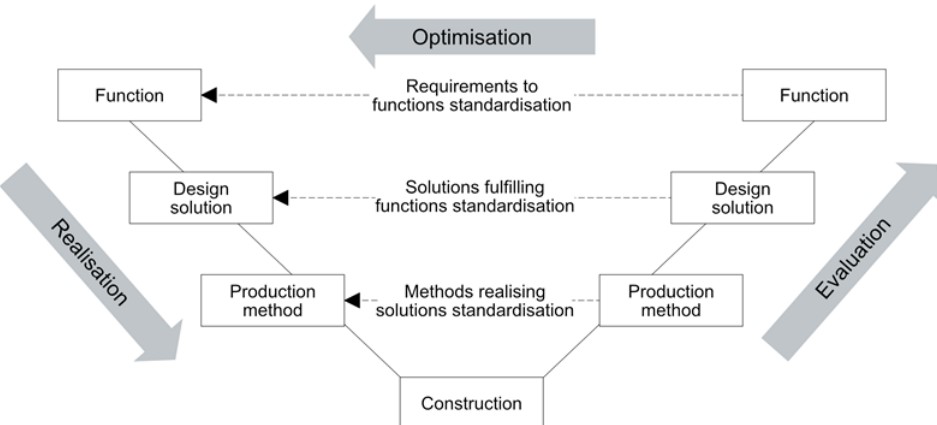

**Figure 1.** The use of configuration information to improve realisation, evaluation and optimisation.

In this study, the selected cases were construction projects delivering MRI rooms that require highly specific configurations. The equipment is vibration sensitive, and the image resolution depends on the dampening of building displacements [25]. Vibration isolation is especially critical during the patient examination of small lesions or long physiological events [26]. Besides vibration damping, the MRI rooms entail other structural challenges, such as increased load capacity due to heavy equipment. There are also requirements on a minimum distance to magnetic steel structures to prevent interference with the equipment, which can affect the image resolution [27]. Daily operations induce disturbing vibrations, such as real-estate systems and footsteps [28]. Evidence suggests that the optimal location of MRI rooms is on the ground floor since the soil has a vibration dampening effect [29]. A slab with increased framing and/or thickness effectively reduces vibration and accommodates the specific requirements [27]. Denser distances between columns may reduce vibrations [30] but limit possible layouts in underlying spaces.

*1.3. Decision Support Systems*

The Architecture, Engineering, Construction and Operation (AECO) industry often focuses on using Building Information Modelling (BIM) data for specific purposes [31]. Decision support systems require data exchange with an end-to-end perspective. AECO processes require collaborations between industries, segments and organisations [32], including cross-cutting integration of technologies [8]. The AECO industry uses IFC schemas to enable data transfer between different actors [33]. This exchange is essential for decisions regarding hospital configurations in construction projects. Decisions based on data from current and previous projects provide opportunities to analyse the effects of different options. However, this requires identifying management models and necessary technologies [34]. Decision support systems contain four main components: user interface, knowledge base, data and model management [35]. The user interface should provide the information required to make decisions. The knowledge base describes operative processes or problems and identifies the data required [36]. Data management includes storage and providing accurate input to processing techniques. Model management describes the objectives and defines relationships to enable data processing [37]. Decision making is complex and quantitative data analyses are not always the best solution since situations and information required may vary. Therefore, decision makers must understand what the decision information represents [38]. A change from event to data-driven decisions requires managerial and cultural differences decision makers must understand within and between companies [39].

*1.4. Contribution to Sustainable Development*

Data-driven processes describe the input and the analyses that enable evidence-based decisions. The digitalisation of configuration information may provide value beyond the

lifecycles of a building by making data accessible across projects and organisations, as shown in Figure 2. Potentially, it can increase the number of informed decisions regarding hospital configurations. Hospital configuration data provide a basis for decisions and contribute to sustainable development in the AECO industry. There are several connections to the UN Sustainability Development Goals [40], as shown in Table 1.

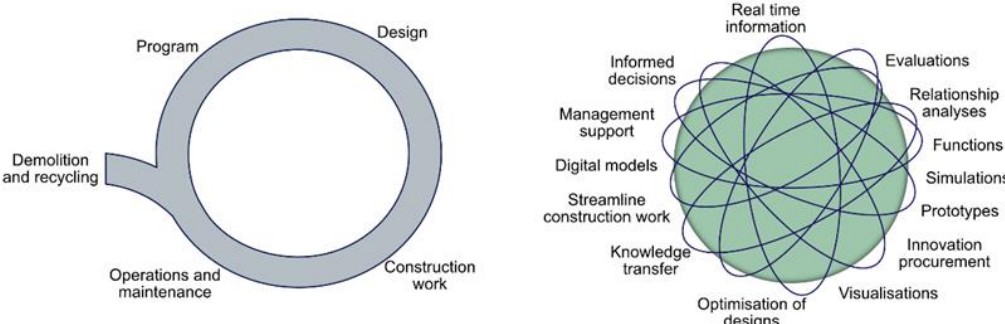

**Figure 2.** Illustration of building information compared to a data lifecycle perspective.

**Table 1.** The connection between UN Sustainability Development Goals [40] and informed configuration decisions based on data.

| Sustainability Development Goals | Connection to Configuration Decisions |
|---|---|
| SDG 3—Ensure healthy lives and promote well-being for all at all ages<br>SDG 11—Make cities and human settlements inclusive, safe, resilient and sustainable | Providing adequate healthcare facilities based on informed configuration decisions enables good medical care for citizens. |
| SDG 6—Ensure availability and sustainable management of water and sanitation for all<br>SDG 7—Ensure access to affordable, reliable, sustainable and modern energy for all<br>SDG 9—Build resilient infrastructure, promote inclusive and sustainable industrialisation and foster innovation<br>SDG 12—Ensure sustainable consumption and production patterns<br>SDG 14—Conserve and sustainably use the oceans, seas and marine resources for sustainable development | As shown in Figure 1, optimisation of hospital configurations may contribute to sustainable multifunctional buildings with reduced resource extraction. |
| SDG 8—Promote sustained, inclusive and sustainable economic growth, full and productive employment and decent work for all | Enable predefined production methods based on configuration information that considers feasibility and working conditions. |
| SDG 17—Strengthen the means of implementation and revitalise the global partnership for sustainable development | Global interdisciplinary research into digitalisation and end-to-end solutions regarding configuration processes may increase value and knowledge. |

## 2. Objective

The management of configuration information is essential for fulfilling the intended hospital functions [5,23]. Inadequate building performance can affect patient diagnosis, treatment, and care [41]. Digital building models contain a large amount of data that can support configuration decision makers [9]. A transparent and accessible understanding of what is to be built, and how this is realised and verified are practical implications of data supported decisions. The objective of this study was to establish a decision support system containing:

- A knowledge base from reviews of literature
- Model management using an ER diagram for a relational database
- Data management by proposing relevant IFC schemas for building model exchange
- A case study investigated gaps in current practice compared to the ER model

The knowledge contributes to supporting hospital configuration decisions during construction projects. The purpose of the study is to answer the following research questions:

1.  Which information is required for hospital configuration decisions?
2.  How are hospital configuration data related?
3.  Which IFC schemas are relevant for configuration information exchange?
4.  What information is available in current construction projects delivering MRI rooms?

## 3. Method

The operational problem from which this research originates was hospital configurations not fulfilling intended functions at the end of construction projects. Configuration information management aims to ensure product functions. A deductive and explorative approach was used for this study to investigate how hospital configuration data can support decisions during construction projects. The method used was based on decision support system research. Four main parts were investigated, as follows:

1.  Problem definition establishes a knowledge base (variables) and model management (variable relationships), while the solving phase defines data management (domain limitations) and user interface (predictions) [42]. The knowledge base was the ISO 10007 guidelines [1] and a literature review, which defined variables required for configuration decisions. This knowledge was compiled in a decision flow chart.
2.  Model management contained variables for configuration decision support presented as an ER model for a relational database describing entities, attributes, and relationships.
3.  The domain limitations for sharing the data were studied. Identification of relevant IFC schemas for each attribute in the ER model ensures data exchange regarding configuration decisions.
4.  The decision interface in this study consisted of empirical investigations of available information for decisions about MRI room configurations compared to the proposed decision system.

The connection between decision support system methods and the four areas of investigation of this study is shown in Figure 3.

Data-driven decisions contain different sources apart from digital building models. This study is limited to configuration decisions based on digital building data with application to hospital construction projects. The purpose of this approach was to improve the possibilities for practical implementations. Testing the ER model with case data was not feasible since much of the required input was missing. Instead, the aim of the case studies on MRI room configuration information was to provide in-depth knowledge of operational gaps. This knowledge provides insights about obstacles to implementing configuration decision support based on building data.

### 3.1. Determine Configuration Decision Flow

The ability to make data-driven decisions rely on a knowledge base of evidence and experience. Defining the decision problem is the first step in decision making analysis [43]. Reviews of the ISO 10007 guidelines for configuration management [1] and construction project research identified the required variables for hospital configuration decisions. A decision flow chart was developed based on the knowledge from the reviews, providing a theoretical ground that validated the choice of variables. Figure 4 shows the Preferred Reporting Items for Systematic Reviews and Meta-Analyses [44] flow from a search in the Elsevier Scopus database to the final selection. The literature selection was limited to recent studies (2018–2021). Manual assessment of the texts was the basis for selecting studies relevant to configuration decisions during construction projects. The word frequency function in MATLAB Text Analytics Toolbox [45] identified common words and the context surrounding those were assessed manually.

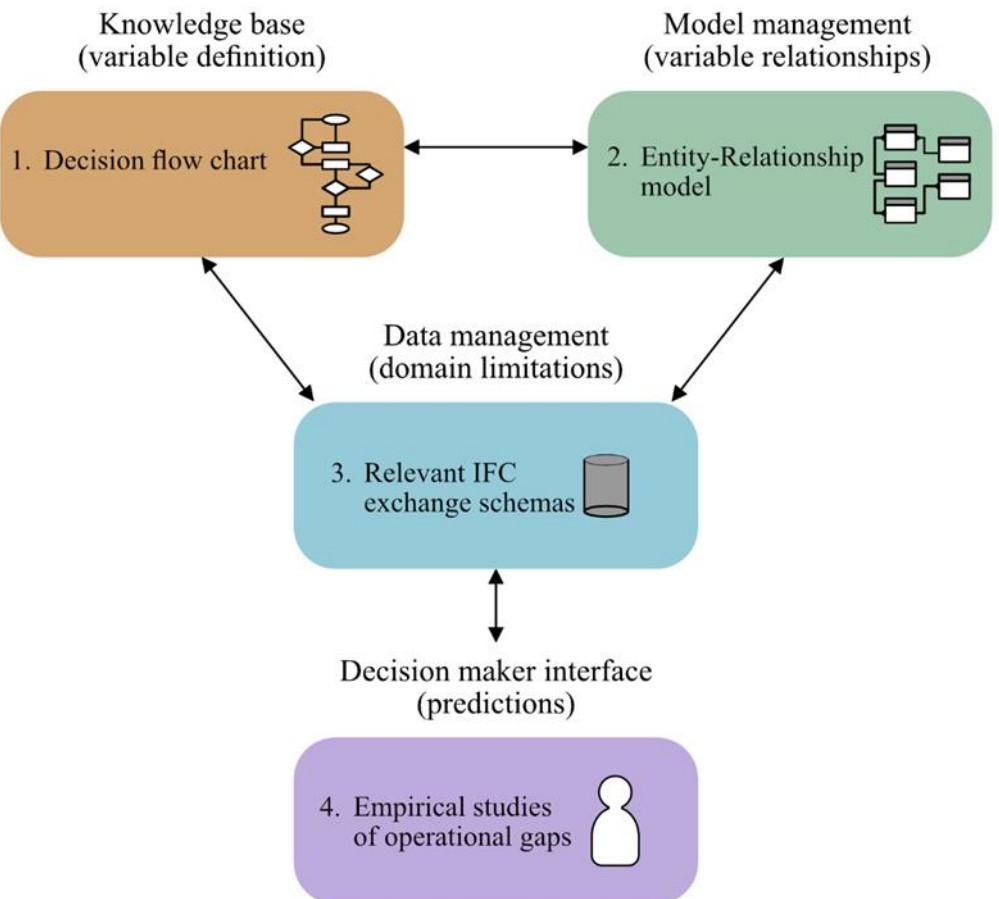

**Figure 3.** Description of the research process. The numbers indicate the order in which the investigations were performed.

### 3.2. Establish an Entity-Relationship Model

Conceptual database ER models establish logical relationships that give new insights into operational issues [46,47]. The design of relational databases requires an ER diagram to establish entities and their relationship. Each entity must have a unique primary key to identify its occurrences. These become a foreign key in another entity if included as an attribute. Composite attributes consist of several sub-attributes that together specify the data required for each entity. Relationships between entities are represented by cardinality, such as one-to-one or one-to-many [48]. The internal validity of ER diagrams consists of theoretically grounded variables and the logical reasoning of their relevance to practice [49,50]. The knowledge base about hospital configuration decisions presented as a flow chart was the basis for selecting the proposed ER model entities and composite attributes. Based on the literature review, the entities were the key areas found and the composite attributes specify the type of data required for decisions. The use of ER models as a research method has limitations since these are restricted to representing complex relationships. However, it meets the purpose of these studies to establish the overall model management connected to the other parts of a decision support system, shown in Figure 3.

### 3.3. Identify Relevant IFC Exchange Schemas

Building models contain data about hospital configurations that can support decisions. Industry Foundation Classes (IFC) schemas are an exchange format for building data and were chosen to provide the proposed ER model with input. The consistency of the ER diagram analysis depends on valid data [47], which the IFC schemas can retrieve from digital building models [33]. For each of the composite attributes in the ER model, relevant IFC schemas that could retrieve data were established. The semantic definition

of the IFC entities constituted the first selection of relevant schemas. Thereafter, the attribute definition for each entity was compared to the results of the literature review. Configuration data can have different sources, but these studies were limited to using IFC schemas since these are the international standard for exchange in the AECO industry. The purpose of this study was to provide a conceptual model; therefore, detailed input data for configuration decisions was not part of this study. The establishment of IFC schemas increases the model's external validity by ensuring the data exchange necessary for configuration decision support.

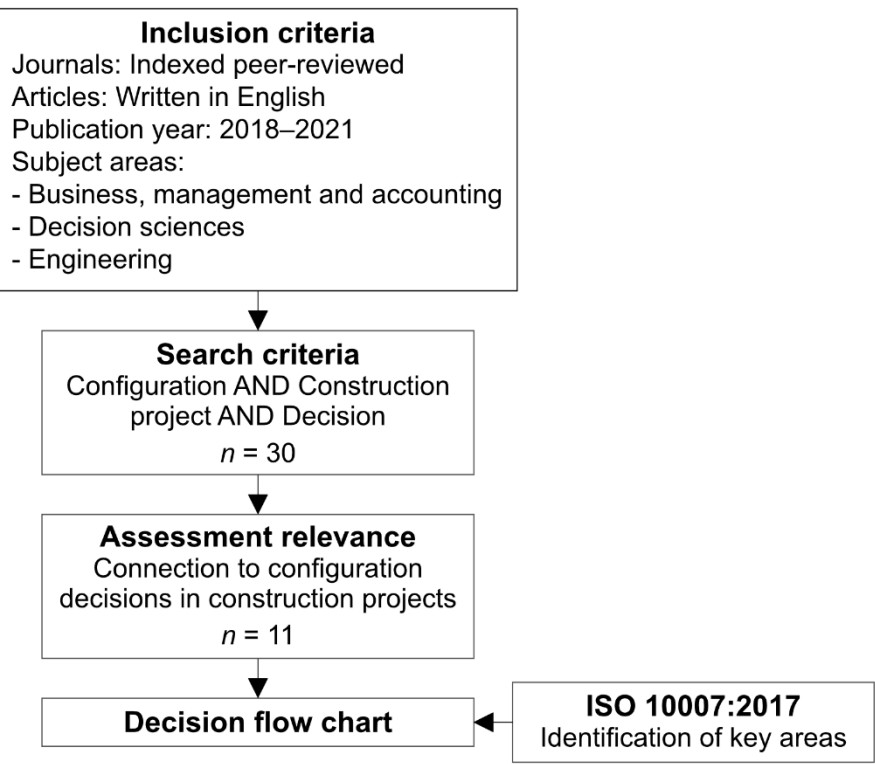

**Figure 4.** The method used for the literature review.

### 3.4. Empirical Studies of Operational Gaps

Case studies are suitable for an in-depth exploration of existing practices in information modelling [49]. A gap analysis compared the IFC schemas of the proposed ER model to the ones used for exchange between AECO actors in construction projects of MRI rooms. The choice was to specifically investigate floor configuration data, since decisions about vibration isolation are essential for equipment performance, which can affect patient diagnosis. The initial selection criteria for the cases were construction projects of new MRI rooms completed from 2015 onwards. Active work with building models during the projects with continuous exchange of building model data between AECO actors was the second selection criterion. Three out of fifteen cases found fulfilled both criteria. Two cases were new builds and one was a new MRI room through a major reconstruction of an existing building. The construction work finished in 2019 (Case 1) and 2017 (Case 2) for the new builds and 2016 (Case 3) for the major reconstruction (initially built in 1975). A limited selection of cases enabled studies of similarities and differences between projects having the same specific delivery [51]. Project documents provided information about the overall configuration information management. All cases used Revit [52] for modelling the building data. Every two weeks, the different models were uploaded using an IFC 2 × 3 Coordination View 2.0 schema to identify the need for adjustments between disciplines.

## 4. Results

A literature review was the basis for the decision flow chart for configuration decisions in construction projects (see Figure 5) and shows the complexity of the processes. Several stakeholders must exchange data to provide accurate information for decisions. Based on the decision flow chart, an ER model for data management was established, describing entities, composite attributes and relationships. It shows how closely connected the entities are and the complexity of attributes required. Identification of relevant IFC schemas shows a potential to use building model data for configuration decisions. However, the case study identifies gaps between the ER model schemas and those used to exchange data in current practice.

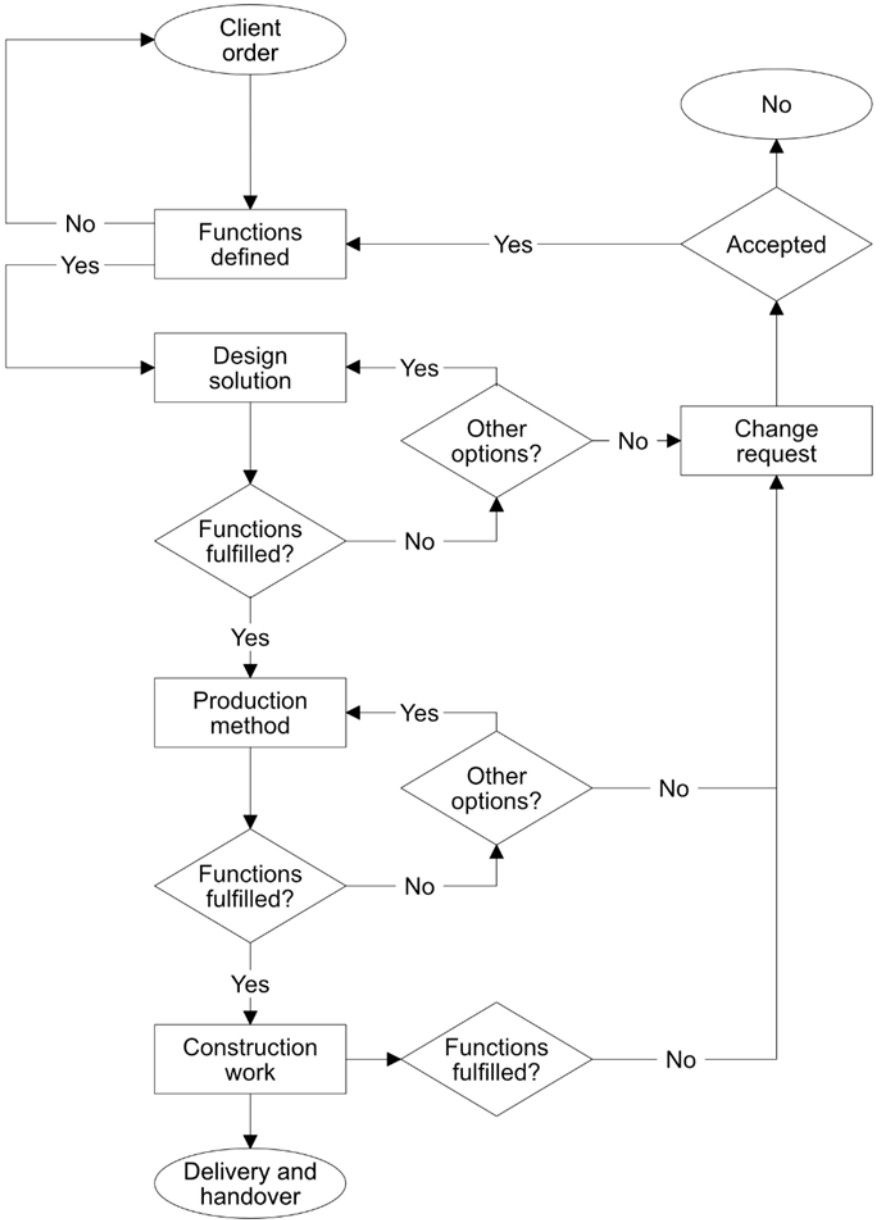

**Figure 5.** Configuration decision flow chart for construction projects.

### 4.1. Decision Flow Chart

Reviews of the ISO 10007 guidelines [1] identified the key areas for configuration decisions. The definition of functions of a product from a lifecycle perspective constitutes the baseline configuration. During design and production, the baseline is the boundary condition for the choice of solutions and methods. For every change to the baseline configuration,

the consequences for the functionality should be analysed. Decided configuration changes become an integrated addition to the baseline. The review of the ISO 10007 guidelines identified five key areas associated with building configurations:

- Functions that constitute the baseline configuration.
- Design of solutions fulfilling the functions.
- Predefinition of production methods that ensure implementation of the solutions.
- Establishing of acceptance values and measurement methods for verification of the baseline configuration.
- Change control to ensure functions according to the intentions of the baseline configuration.

The flowchart for construction projects describes the decision processes associated with the five key areas (see Figure 5). The first decision is the client order with required functions, including acceptance values (i.e., the baseline configuration). If all functions are not adequately defined, the client must clarify these and decide how they should be prioritised. Thereafter, the decisions regarding the choice of design solutions are verified against the baseline functions with predefined measurement methods. Alternative solutions should be explored if a function is not fulfilled. Unattainable functions should render a change request and consequence analyses, after which the client decides whether or not it should be implemented. All accepted changes start the process from the beginning with the definition of functions. The decision about production methods should be verified against the functional acceptance values. If those are not fulfilled and an optional method cannot be found, a change request should be issued. The literature review results regarding configuration decisions during construction projects are shown in Table 2.

**Table 2.** The configuration decision key areas that the found literature addresses.

| References | Decisions | | | | |
|:---:|:---:|:---:|:---:|:---:|:---:|
| | **Change** | **Design** | **Function** | **Production** | **Verification** |
| [53] | x | x | – | x | x |
| [54] | x | – | x | x | – |
| [55] | x | x | x | x | – |
| [56] | x | x | x | x | x |
| [57] | x | x | x | x | x |
| [58] | x | x | x | x | x |
| [59] | – | x | x | x | – |
| [60] | x | x | x | x | x |
| [61] | x | x | x | x | – |
| [62] | x | x | x | x | – |
| [63] | x | x | x | x | – |

x = Addressed; - = Not addressed.

All eleven of the included journal papers mention configuration decisions concerning the choice of options during construction projects. According to the studies, configuration information optimises option evaluations and provides a basis for informed decisions. The provision of measurable functions was mentioned as a condition for supporting configuration decisions in the included studies. Configuration feasibility was addressed in seven studies regarding decisions about production methods, processes and resources. According to ten studies, transferring information from one project to another facilitates more efficient and reliable configuration decisions. Optimisation of configuration decisions requires the development of algorithms that automate options, as mentioned in ten papers. The effect of configuration decisions on sustainability was present in all of the eleven included studies. Two thirds of the papers considered that configuration decisions based on data have several positive effects, such as reduced material and energy consumption and overall project resources (i.e., cost and time).

## 4.2. Entity-Relationship Model

The baseline entity in the ER diagram should contain data necessary for fulfilling required functions (Function ID) and constitute the boundary conditions as a foreign key in all the other entities, as shown in Figure 6. Data about the decision maker (Decision Maker ID) regarding hospital configurations are part of the baseline entity. Decisions about design solution choices (Solution ID) are based upon comparing acceptance values (Acceptance Value ID) for the baseline configuration (Baseline ID), including implemented changes requests (Change Request ID) with solution performance values. The change request entity contains the current baseline, solution and production method against which the consequence analysis (Change Consequence ID) is compared. The client's decision regarding implementation (Implementation Decision ID) or not is also part of this entity. Data about decided verification methods provide directions on how to ensure the fulfilment of functions. Connecting functions and solutions can be the basis for standardisation and optimisations, as shown in Figure 1. Decisions about production methods (Production Method ID) require the same composite attributes as design solutions. The design solution and production method composite attributes are a foreign key to the other entity to ensure feasible decisions that fulfil adequate functions at delivery to the client. This enables decisions that streamline operations throughout the construction phases. Early identification of verification methods (Verification Method ID) for designers and contractors ensures that the decisions about solutions and production methods do not cause deviations from intended functions. The cardinality between entities shows that a baseline has several verification methods and design solutions. A design solution may require several production methods to be realised. The consequences of a change might affect several functional attributes of the baseline. The ER diagram shows that all entities are closely related since the primary keys (PK) are often the foreign key (FK) in another.

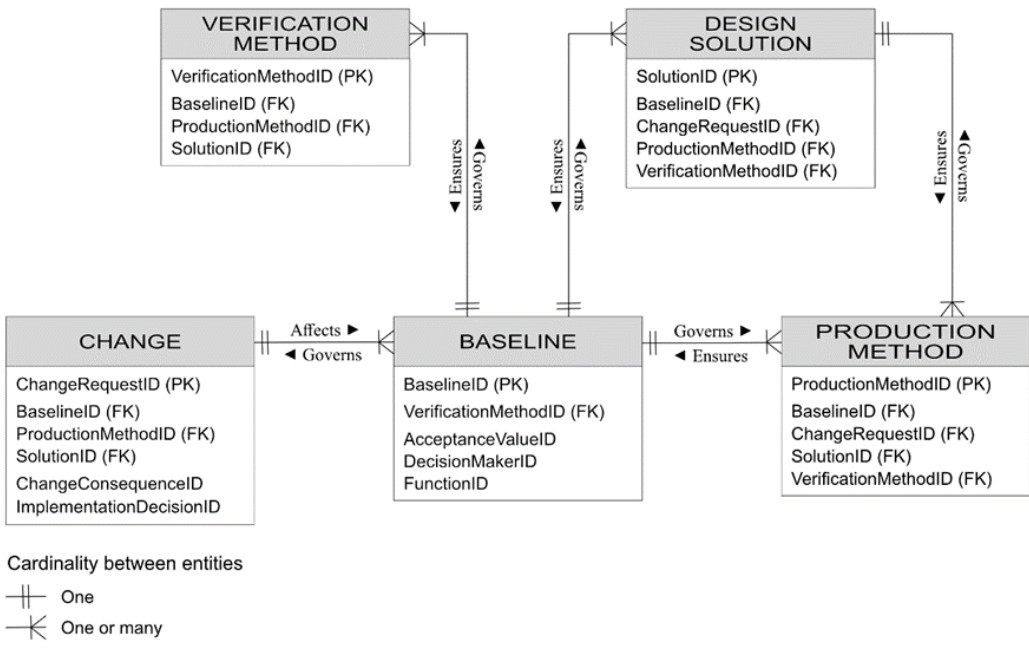

**Figure 6.** The proposed ER model with entities and composite attributes.

## 4.3. Proposed IFC Exchange Schemas

Table 3 shows proposed IFC schemas [33] for data exchange based on the entities in the ER model. Functions govern all configuration decisions and *IfcControl* schema can exchange data about scope boundaries, such as acceptance values. Relations between information associated with the functions can be retrieved using the *IfcRelAssociates*. The use of *IfcPropertySet* provides a container for data describing properties for required functions. The *IfcActor* schema exchange data about stakeholders during the project life cycle,

providing information on the decision maker. Design solutions require exchange from several schemas. Data regarding groups of objects fulfilling a function and the placement of elements are exchanged with *IfcGroup* and *IfcProduct*, respectively. Independent information about physical parts of an object is exchanged with *IfcObject*, which might contain references to other data sources. The *IfcRelDecomposes* schema exchange data regarding relationships between a system and all constituent parts fulfilling one or several functions. Data describing the bi-directional relationship among the objects required for a function is exchanged with *IfcRelAssigns*. The relationship between objects for a function criterion is exchanged with *IfcRelConnects*. External sources of information can be retrieved using *IfcRelAssociates*. The schemas containing data about relationships (*IfcRel*) give important information for the choice of production methods. Retrieving data with *IfcProcess* and *IfcResource* identify work packages and resources to plan the realisation of the baseline configuration. Change requests require information about which objects the new function contains, which *IfcGroup* can obtain. The *IfcRelDefines* schema can provide the consequence analysis with information about which objects share the same properties. The *IfcControl* schema exchanges data about change implementation decisions and verification values by defining their limitations.

**Table 3.** The proposed IFC schemas for each attribute in the ER model and the ones used for exchanging flooring data in the studied cases.

| Proposed IFC Schemas | Database Attributes | | | | | | | | |
|---|---|---|---|---|---|---|---|---|---|
| | Acceptance Value ID | Decision Maker ID | Function ID | Solution ID | Production Method ID | Change Request ID | Change Consequence ID | Implementation Decision ID | Verification Method ID |
| IfcObjectDefinition | | | | | | | | | |
| *IfcActor* | - | x | - | - | - | - | - | - | - |
| *IfcControl* | x | - | x | - | - | - | - | x | x |
| *IfcGroup* | - | - | - | x * | - | x * | - | - | - |
| *IfcObject* | - | - | x * | - | - | - | - | - | - |
| *IfcProduct* | - | - | - | x * | - | - | - | - | - |
| *IfcProcess* | - | - | - | - | x | - | - | - | - |
| *IfcResource* | - | - | - | - | x | - | - | - | - |
| IfcRelationship | | | | | | | | | |
| *IfcRelAssigns* | - | - | - | x | x | - | - | - | - |
| *IfcRelAssociates* | - | - | x | x | x | - | - | - | - |
| *IfcRelConnects* | - | - | - | x ** | - | - | - | - | - |
| *IfcRelDecomposes* | - | - | - | - | - | - | x | - | - |
| *IfcRelDefines* | - | - | - | x | x | - | - | - | - |
| IfcPropertyDefinition | | | | | | | | | |
| *IfcPropertySet* | - | - | x | - | - | - | - | - | - |

x = Exchange schema required for attribute; - = Not applicable; * = Data exchanged in Case 1, 2 and 3; ** = Data exchanged in Case 1.

### 4.4. Operational Gaps

The case study investigated the IFC schemas used in three MRI facility construction projects to exchange data compared to those of the proposed decision support system. None of the projects used building model data as a basis for their decisions. Instead, the decision making was event-driven and based on manually managed logs. There was no real-time update of the building models; the actors did parallel work on the model and uploaded their parts once every two weeks. Hence, the IFC schemas were not used to exchange configuration data in real time during the construction projects. The results show that several IFC schemas related to MRI floor configuration information were not used in the cases, as shown in Table 3. All IFC files applied the schemas; *IfcGroup*, *IfcObject* and *IfcProduct*, containing data about the physical elements of the flooring and their placement but not dependencies between them. This provides part of the information change requests, functions and solutions. The lack of information about dependencies does not provide a sufficient basis for floor configuration decisions, in which relationships between elements and systems are essential. One case also used the *IfcRelConnects* schema for exchanging data about connections between structural elements, which are part of the information required for decisions related to choosing solutions for dampening vibrations (see Table 3). Overall,

the projects did not exchange data sufficient to support floor configuration decisions compared to the proposed ER diagram.

## 5. Discussion

Digital building models can provide accurate data for decision support systems to process and provide valuable information. Building information models contain data that contribute to informed decisions regarding hospital configurations. The proposed decision support system provides conditions for accurate information to assure intended healthcare functions in construction projects. The ER model and relevant IFC schemas ensure data exchange to give information required for configuration decisions. The gap analysis of current practice in the studied MRI construction projects shows that a small part of the data required in the proposed model was exchanged. Digitalisation and automation of processes provide and require accurate configuration data to support decisions. Technologies providing physical-digital interfaces, such as building scanners, contribute with data for configuration decisions [64]. Networks, such as 5G and cloud computing, can transfer data for configuration decisions in real time [65]. Clearly defined and updated baseline configuration information is a prerequisite for enabling digital-physical technology, such as additive manufacturing [21] or robotics [66].

The decision flowchart and ER model provided by this study (see Figure 6) display the complexity of configuration information. It also shows that the current IFC schemas provide the possibility of controlling configuration by providing exchange possibilities (see Table 3). Data about components and systems were exchanged with IFC schemas in the studied cases and connections between structural elements were present in one case. However, the schemas related to other relationships, actors, processes and resources were not (see Table 3). Evaluating which design solutions fulfil certain hospital functions can support client decisions regarding the baseline configuration [5]. Determination of construction methods for each design solution through cross-cutting collaborations facilitates implementation during construction. Hence, the contractors can focus on planning resources and processes instead of incremental method development during production [14]. Verification values for the designers and contractors can be based on data regarding components, systems, and properties. The consequence analysis of configuration changes may be performed using relationship data from digital building models [62]. However, the cases did not provide access to these data to support the client's decision regarding if to implement changes or not.

The benefits of the proposed decision support system are a knowledgebase for developing processes, data- and model management for providing accurate information and availability by identifying current exchange options. The complexity of hospital configurations makes manual information management very difficult and data-based decisions especially beneficial. Probable implications are more evidence-based decisions, knowledge transfer between projects and sustainable development contributions. In addition, preventing corrective rework due to deviations from intended functions reduces the AECO industry's environmental impact and resource extraction [67]. Data supported configuration decisions ensure the delivery of hospital buildings with adequate functionality [5].

The research approach in this study used methods from decision support system research applied to hospital configurations (see Figure 3), of which different parts have limitations. The theoretical knowledge base was restricted to previous studies of configuration decisions in construction projects and the ISO 10007 guidelines [1]. An ER model established the overall database structure to manage data, but lacks details. Data management was limited to IFC schemas and building model data, while other sources were not included. The operational gap analysis in this study consisted of a limited case selection. However, the purpose was to obtain in-depth knowledge of configuration information management in hospital projects with a focus on MRI room flooring.

The proposed decision support system enables standardisation based on data analyses of connections between functions, solutions and construction methods that achieve

intended functions. These connections can be used for future projects to streamline processes and ensure the delivery of hospital buildings with adequate end-user functionality. Decision support systems enable knowledge transfer of data between organisations and countries to optimise hospital configurations for specific functions [2]. The establishment of a baseline configuration with implemented changes is essential for verification and choice of solutions and production methods during the construction project process [7].

## 6. Conclusions

The purpose of this study was to establish a decision support system for hospital configuration during construction projects, using an end-to-end perspective. The flow chart gives knowledge about configuration decisions that can be used to improve processes in construction projects. Definition and verification of required functions are central, which is also shown as the baseline attribute is part of all entities in the ER model. The results show that the proposed decision support model using building model data is feasible. However, the case study identified gaps mainly regarding the data exchange regarding building part relationships and processes. The proposed decision information system shows how data can enable baseline establishment, connections with solutions and production methods and verification throughout the projects. The practical implications are more evidence-based decisions and delivery of hospital buildings with adequate functions. A recommended next step in this research is establishing detailed attributes that support configuration decisions and how to retrieve the input data.

**Author Contributions:** Conceptualization, P.S., M.L. and A.A.; methodology, P.S.; software, P.S.; validation, P.S.; formal analysis, P.S.; investigation, P.S.; data curation, P.S.; writing—original draft preparation, P.S.; writing—review and editing, P.S., M.L. and A.A.; visualization, P.S.; supervision, M.L. and A.A.; project administration, P.S., M.L. and A.A. All authors have read and agreed to the published version of the manuscript.

**Funding:** Region Stockholm funded this research (grant number LDR6262). KTH Royal Institute of Technology funded the APC.

**Data Availability Statement:** The case data from this study are available from the authors upon request.

**Acknowledgments:** A grant from Region Stockholm (previous Stockholm County Council) enabled this research, for which the authors are most grateful.

**Conflicts of Interest:** The authors declare no conflict of interest. The funders had no role in the design of the study; in the collection, analyses, or interpretation of data; in the writing of the manuscript; or in the decision to publish the results.

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
