# Peer review of "A Decision Support System for Hospital Configurations in Construction Projects"

_buildings, doi:10.3390/buildings12101569_

Round 1
Reviewer 1 Report
This paper(buildings-1908329) seeks to develop a decision support system by establishing a value chain of configuration information with an end-to-end perspective. The reported study may be suitable for the journal; however, it needs a considerable amount of work or revision before it fulfills the standard requirements of the journal.
1) [page 10, figure 5] The flow chart needs to be further processed to make the logic more clear.
2) [section 4.2] The section 4.2 should be present in a more structural way. There are many illustrations of the relationships between entities in figure 6, and it is recommended to highlight the key connections.
3) For the tables, the authors should add the notes corresponded to each special symbol, such as “-” and “x” in table 2, to facilitate the reader's understanding.
4) The paper uses many tables to present the final results. It is suggested that richer visualizations can be used in the results and discussion sections of the paper.
5) [page 13-14, section 5] The part of discussion seems quite weak about the result.
Author Response
Dear reviewer,
Thank you for reviewing and providing valuable comments on our manuscript. Changes have been performed to the text and references where appropriate. Please see the attachment where the changes are shown with red text. According to your suggestions, the following alterations are done:
- Figure 5 is revised, as is the adjacent text.
- Section 4.2 is structured to be more similar to the ER model. The text is altered to clarify key connections.
- Table footnotes are added to explain symbols.
- Tables 3 and 4 are merged into one.
- The discussion section is revised to more clearly describe the results.
Sincerely regards
The authors

Reviewer 2 Report
The manuscript “A decision support system for hospital configurations in construction projects” is interesting. However, there are some weaknesses in the manuscript.
1. Evidence from past research to support the knowledge is lacking in the paper.
2. The authors should summarize the research process methodology in a figure.
3. The conclusion section should highlight the major contributions of the study to knowledge.
4. The study should provide the practical implications of the study.
5. The should highlight the implications and scope of the study.
Author Response
Dear reviewer,
Thank you for a thorough and constructive review of our manuscript. Changes have been performed to the text and references where appropriate. Please see the attachment where the changes are shown with red text. According to your suggestions, the following alterations are done:
- The relevance of the references is reviewed and clarified in the text
- The text regarding research design and methods is revised.
- Figure 3, which describes the research method process is altered, as is the adjacent text.
- The conclusion section is revised to highlight contributions.
- Implications and scope are clarified in the introduction, discussion and conclusions.
Sincerely regards
The authors

Round 2
Reviewer 2 Report
The paper has been improved.